# Cardiovascular health of women 10 to 20 years after placenta-related pregnancy diseases considering the possible effect of pentaerythrityl tetranitrate treatment during pregnancy on long-term maternal cardiovascular health (PAVA study)

**Charlotte Lößner**[1][☯]**, Anna Multhaup**[1][☯]**, Laura Bäz**[2]**, Thomas Lehmann**[3]**, Yvonne Heimann**[1]**, Ekkehard Schleußner**[1]**, Marcus Franz**[2]**, Tanja Groten**[1] *

1 Department of Obstetrics, University Hospital, Friedrich Schiller University, Jena, Germany, 2 Department of Internal Medicine 1, University Hospital, Friedrich Schiller University, Jena, Germany, 3 Department of Medical Statistics, Information Technology und Documentation, University Hospital, Friedrich Schiller University, Jena, Germany

☯ These authors contributed equally to this work.
* tanja.groten@med-uni-jena.de

## Abstract

### Background

Women developing preeclampsia (PE) or fetal growth restriction (FGR) during pregnancy are at higher risk for cardiovascular diseases (CVD) later in life. We aimed to analyse cardiovascular health of women 10–20 years after affected pregnancies in comparison to women after uneventful pregnancies. In addition, we assessed a potential long-term effect of the NO-donor pentaerythrityl tetranitrate (PETN).

### Methods

Women 10–20 years after severe PE, including women receiving PETN during pregnancy and matched controls were recruited and assessed for baseline clinical data and cardiovascular function by transthoracic echocardiography, VICORDER and USCOM. SPSS was used for statistical analysis.

### Results

53 participants after PE/FGR (13 with former PETN intake) and 51 controls were recruited for follow-up at an average of 14 years after index pregnancies. Compared to controls, women after PE/FGR had a significantly higher incidence of arterial hypertension (13.7% vs. 41.5%, p<0.001), and were more likely to be hypertensive (41.2% vs. 67.30%, p = 0.008). There were no differences in cardiovascular function observed. Affected women

**Data Availability Statement:** All relevant data are within the manuscript and its Supporting Information files.

**Funding:** This work was funded by the Deutsche Forschungsgemeinschaft (DFG, German Research Foundation) Clinician Scientist Program OrganAge funding number 413668513 and by the Interdisciplinary Center of Clinical Research of the Medical Faculty Jena to AM.

**Competing interests:** The authors have declared that no competing interests exist.

with PETN intake during pregnancy showed lower mean values for right atrial area and ventricle in comparison to controls and also to affected women without former medication.

## Conclusions

In conclusion, our study results confirm that the risk of CVD is increased in women after PE/FGR compared to women after uneventful pregnancies. Contrary to our expectations, no major cardiovascular changes were observed in our cohort 10–20 years post pregnancy. The observed differences found in right heart dimensions were within reference ranges, and should be interpreted with caution.

## Introduction

Preeclampsia (PE) and fetal growth restriction (FGR) are placenta-associated pregnancy complications characterized by maternal endothelial dysfunction leading to malperfusion of maternal organs such as liver, brain, kidneys, and/or the placenta itself. FGR occurs in up to 10% of all pregnancies [1], whereas 2 to 8% of pregnancies are complicated by PE [2]. Current hypothesis on the etiology proposes that defective remodeling of endometrial spiral arteries causes placental hypoperfusion and hypoxia. Resulting oxidative stress is thought to trigger an excessive systemic inflammatory response causing maternal endothelial dysfunction and vasoconstriction. Both pathologies are characterized by an anti-angiogenic growth factor profile, reflecting the activated state of maternal endothelium. Women affected, show reduced levels of endothelial-protective growth factors like vascular endothelial growth factor (VEGF) and placental growth factor (PlGF), whereas levels of endothelial activators such as soluble fms-like tyrosine kinase-1 (sFlt-1) and soluble endoglin (sEng) are elevated [3]. Women experiencing pregnancies accompanied by such antiangiogenic conditions are known to be at risk for later cardiovascular disease (CVD). A large retrospective cohort analysis by Smith et al. demonstrated an increased risk of ischemic heart disease over a follow-up period of 15–19 years [4], which was confirmed in a recent study demonstrating that risk of subsequent maternal heart failure was increased after hypertensive pregnancy complications [5].

Recently, our group published that the NO donor pentaerythrityl tetranitrate (PETN), when administered to pregnant women at risk for developing PE and/or FGR (identifiable by impaired uterine perfusion in midpregnancy), can improve pregnancy outcome [6, 7]. In a prospective, randomized, placebo-controlled, multicenter trial, PETN significantly improved maternal outcome by reducing the incidence of pregnancy-related hypertension and PE, suggesting an impact of PETN on maternal vascular health [6]. We suspect that this effect is not solely due to its vasodilatory function, as a protective effect on endothelial function has already been demonstrated in clinical trials in cardiovascular patients [8]. However, a possible long-term effect of PETN on cardiovascular function in affected pregnant women has not yet been investigated.

Clinical changes in women after placenta-related pregnancy complications have been examined in several studies. There is evidence that arterial stiffness remains elevated in women after PE [9]. Arterial stiffness is considered an independent predictor of cardiovascular morbidity and early cardiovascular mortality and can be quantified by pulse wave velocity (PWV) or augmentation index (Aix) [10]. Furthermore, a raised total vascular resistance (TVR) proved to be the best single predictive factor for the development of arterial hypertension in and after pregnancy [11]. Women with a history of hypertensive pregnancy disorders

continued to have elevated values long after delivery compared to controls [12]. In addition, women after pregnancies complicated by PE showed persistent microcirculatory dysfunction, as substantiated by markedly decreased flow-mediated dilation (FMD) values [10]. Similar results were obtained in follow-up studies of women after pregnancies complicated by FGR [13]. Further techniques to evaluate cardiovascular and endothelial function have been established. Transthoracic Echocardiography (TTE) is the most common procedure for evaluating cardiac structure and function. The USCOM doppler monitor is based on continuous-wave doppler signal analysis, measuring transvalvular ventriculoarterial aortic or pulmonary blood flow and has recently been validated for monitoring of hemodynamics in and out of pregnancy [14]. Via VICORDER, indices of large artery elasticity and vascular tone regulation, including arterial stiffness, arterial-wave reflection and endothelial function can be analysed using cuff-based oscillometry.

The aim of this study was to analyse cardiovascular health of women 10–20 years after pregnancies complicated by PE and/or FGR, in comparison to women with uneventful pregnancies. In addition, we intended to assess a potential protective effect of PETN taken during pregnancy on later cardiovascular function.

## Methods

### Design and study population

This prospective, cross-sectional, single-center trial was conducted at the Department of Obstetrics at the University Hospital in Jena, Germany. Women who developed PE during their pregnancies and delivered at our institution during 2003–2009 between 24 and 42 weeks of gestation were identified by the ICD for PE (O14) through our hospital database. According to guidelines from International Society for the Study of Hypertension in Pregnancy (ISSHP), PE was defined as the occurrence of blood pressure exceeding systolic blood pressure of 140 mmHg and/or diastolic blood pressure of 90 mmHg and coexisting proteinuria exceeding 300mg/24h. After exclusion of multifetal pregnancies and those affected by fetal malformations, patient records were reviewed to select cases of particularly severe PE. Severe PE was defined as onset before 34 weeks of gestation and/or blood pressure exceeding 160 mmHg systolic and/or 110 mmHg diastolic. Additionally, we recruited women following pregnancies with intake of PETN. Eligible were participants from the PETN-pilot study who were recruited because of an abnormal uterine artery Doppler at mid-trimester scan either receiving PETN or Placebo [7] and women who received PETN as a personalized therapy attempt between 2010 and 2015 in cases of PE with accompanied FGR. Controls who experienced healthy pregnancies were chosen according to following matching criteria: year of delivery, weeks of gestation, maternal age at delivery and number of previous pregnancies and births. Exclusion criteria for enrollment were: multifetal pregnancies, any fetal anomalies, existence of any maternal disease or circumstance causing a high-risk pregnancy. All identified women were invited to study participation by mail. Consenting participants were examined at our study clinic from October 2019 to December 2021. Written informed consent was obtained for each enrolled patient. Data handling accorded to the European data safety regulations. Data capture was performed using paper based clinical recording forms. In order to ensure a pseudonym analysis of data, each patient data set was given a unique patient identification number when being entered into the study database. Data of index pregnancies were retrieved from hospital database. Ethical approval was obtained by the ethical committee of Friedrich-Schiller-University in Jena (no.: 2019–1498_BO). The study is registered at U.S. National Library of Medicine Clinical-Trials.gov (no.: NCT04484766).

## Clinical characteristics

Clinical data on consumption behaviour, education level, current medication intake and comorbidity were collected using a standardized questionnaire. Variables on index pregnancy were retrieved from our hospital-based information systems, obstetrical charts and, in part, on information provided by participants. Height and weight were measured using standardized measuring and calibrated personal scales. BMI was then calculated according to the internationally recognized formula. Preexisting conditions were defined by participant's report of physician's diagnosis and use of medication. Blood pressure was registered from non-invasive measurements using automated oscillometric devices. Physical activity was assessed by using International Physical Activity Questionnaire (IPAQ) and categorized according to published guidelines [15]. Body composition including body fat und muscle mass was determined by Bioelectrical Impendance Analysis (BIA) [16]. Carotid intima-media thickness (CIMT) was measured to diagnose the extent of atherosclerotic changes. We therefore measured thickness of the two inner layers of carotid artery by ultrasound according to specifications from conventional study protocol [17]. Blood samples of basic serum analysis were collected, processed and analyzed according to standard protocol at Department of clinical chemistry at the University Hospital Jena.

## Transthoracic echocardiography

Transthoracic Echocardiography (TTE) was performed by two experienced cardiologists using a Phillips Epic 7c ultrasound machine (Philips Healthcare, Andover, MA, USA). All patients underwent complete TTE study including 2D, M-Mode, color flow and spectral doppler. All measurements were performed according to standards of the European Association of Echocardiography [18]. All images were initially digitally stored for later off-line analysis using the software Image-Arena™ Version 4.6 (TomTec Imaging Systems, Unterschleissheim, Germany).

## Haemodynamics

USCOM analysis (USCOM Ltd., Sydney, Australia) and VICORDER measurement (Skidmore Medical Ltd, Bristol, United Kingdom) were performed at study visit by trained study physicians, according to manufacturer's instructions. For both measurements, women were positioned supine with upper body bent to 30˚. Patients were not allowed to speak or move during examination. Brachial BP was entered from manual measurements taken. Using USCOM, we measured aortic blood flow by positioning the doppler probe at jugular fossa. Quality of determined signal was checked by typical acoustic Doppler signal and by graphical representation of velocity-time integral (VTI) on USCOM monitor. For VICORDER analysis we determined Pulse Wave Velocity (PWV), Pulse Wave Analysis (PWA) and Flow Mediated Slowing (FMS) as described by manufacturer. FMS analysis was performed using variant with upper-arm and wrist occlusion. A vacuum cushion was added for safe positioning of the arm.

## Statistical analysis

Categorical variables were compared between groups by Fisher's exact test or Chi-squared test, and absolute and relative frequencies are reported in each group. Non-parametric Mann-Whitney U Test was applied to compare continuous variables, and median as well as $25^{th}/75^{th}$ percentile were provided to summarize the data. All tests were two-sided. Results were considered statistically significant at $p < 0.05$. Statistical analyses were performed with IBM SPSS Statistics version 28.0 (IBM Corp., Armonk, N.Y., USA).

## Results

Between 2003 and 2009, we identified 416 women with a history of PE who gave birth at University Hospital Jena. Of these, 88 met inclusion criteria and were invited to participate in our study. In addition, all 110 participants from PETN pilot study were invited. Of 22 cases who received PETN as a personal healing attempt between 2010 and 2015, five were invited as they met inclusion criteria. Finally, the study cohort consisted of 53 women after PE/FGR and 51 women with a history of uneventful pregnancies. Fig 1 provides an overview of the enrolment of the PAVA study population (Fig 1).

Clinical characteristics comparing women after uneventful pregnancies to women with a history of PE/FGR are displayed in Table 1. Current cardiovascular evaluation took place on average 14 years after index pregnancies (p = 0.448). Women post-PE/FGR had a significantly higher overall incidence of chronic diseases (p<0.001), with significantly higher rates of arterial hypertension (p<0.001) and need for regular antihypertensive medication (p = 0.002). Significantly more women after PE/FGR were hypertensive at study evaluation (p = 0.008), with blood pressure values exceeding 140 mmHg systolic and/or 90 mmHg diastolic. While BMI did not differ between groups, results of BIA showed a significantly higher mean body fat mass in women with previous PE/FGR (p = 0.026) (Table 1).

Group comparison of cardiovascular function analysis performed by TTE showed a larger mean left ventricular posterior wall diameter (LVPWD) in women after PE/FGR (p = 0.036), whereas USCOM and VICORDER analysis did not reveal significant differences (Table 2).

Table 3 compares clinical characteristics of 13 women after PE/FGR with PETN use during pregnancy to 40 previously affected unmedicated women. There were no significant differences between the two groups, except for current regular medication intake, which was significantly lower in women with former PETN intake (p = 0.034). In addition, serum analysis revealed significantly lower levels of both total cholesterol (p = 0.044) and LDL cholesterol (p = 0.028) in women using PETN during affected pregnancies (Table 3).

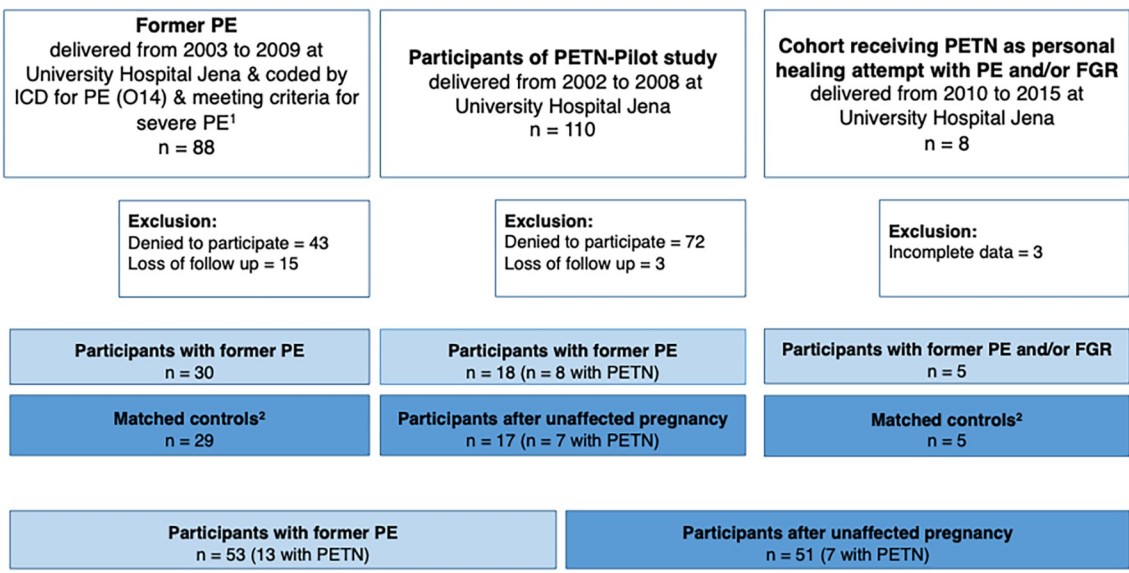

**Fig 1. Outline of enrollment of PAVA-study population.** [1]severe PE: defined as onset before 34 weeks of gestation and/or blood pressure exceeding 160 mmHg systolic and/or 110 mmHg diastolic; [2] matching criteria: year of delivery, weeks of gestation, maternal age at delivery and number of previous pregnancies and births.

**Table 1. Group characteristics\*.**

| | Women with uneventful pregnancies (N = 51) | Women with former PE/FGR (N = 53) | p\*\* |
|---|---|---|---|
| **Characteristics at index pregnancy** | | | |
| Maternal Age at delivery[(years)] | 32 (28–35) | 31 (26.5–34.5) | 0.341 |
| Gestational age at delivery[(weeks of pregnancy)] | 39 (38–40) | 34 (28–36.5) | **<0.001** |
| Birth Weight percentile child[(%)] | 41 (29–65) | 6 (2–18) | **<0.001** |
| Presentation of pregnancy complication | | | |
| PE | | 23 (43.4%) | |
| FGR | | 13 (24.5%) | |
| PE and FGR | | 17 (32.1%) | |
| Chronic diseases during index pregnancy | 4 (7.8%) | 32 (6.4%) | **<0.001** |
| Diabetes mellitus | 0 (0%) | 1 (1.9%) | 0.327 |
| Arterial hypertension | 2 (3.9%) | 29 (54.7%) | **<0.001** |
| Thyroid Diseases | 0 (0%) | 1 (1.9%) | 0.327 |
| Other Diseases | 3 (5.9%) | 5 (9.4%) | 0.499 |
| Medication intake during index pregnancy | 11 (21.6%) | 46 (86.8%) | **<0.001** |
| PETN | 7 (13.7%) | 13 (24.5%) | 0.164 |
| Antihypertensive medication | 1 (2%) | 12 (22.6%) | **0.002** |
| Aspirin | 0 (0%) | 5 (9.4%) | **0.025** |
| Other medication[1] | 4 (7.8%) | 22 (41.5%) | **<0.001** |
| **Characteristics at evaluation** | | | |
| Time since index pregnancy[(years)] | 14 (13–16) | 14 (12–16) | 0.448 |
| Age[(years)] | 45 (42–49) | 44 (40.5–47.5) | 0.163 |
| BMI[(kg/m2)] | 24.8 (22.72–29.49) | 27.29 (22.95–32.81) | 0.138 |
| BP systolic[(mmHg)] | 133 (124–146.5) | 137 (124–149.9)[N = 52] | 0.337 |
| BP diastolic[(mmHg)] | 87 (78.5–94) | 89.5 (81.8–96.9)[N = 52] | 0.190 |
| Hypertension[2] | 21 (41.2%) | 35 (67.3%)[N = 52] | **0.008** |
| Chronic diseases | 17 (33.3%) | 36 (67.9%) | **<0.001** |
| Diabetes mellitus | 1 (2%) | 2 (3.8%) | 0.583 |
| Arterial hypertension | 7 (13.7%) | 22 (41.5%) | **0.002** |
| Thyroid Diseases | 4 (7.8%) | 12 (22.6%) | **0.037** |
| Lipid Metabolism Diseases | 1 (2%) | 1 (1.9%) | 0.978 |
| On Medication | 20 (39.2%) | 37 (69.8%) | **0.002** |
| Antihypertensive medication | 7 (13.7%) | 22 (41.5%) | **0.002** |
| Other medication[1] | 18 (35.3%) | 28 (52.8%) | 0.087 |
| Consumption behaviour | | | |
| Smoking | 9 (17.6%) | 10 (18.9%) | 0.359 |
| Alcohol consumption | 39 (76.5%) | 38 (71.7%) | 0.581 |
| Drug consumption | 3 (5.9%) | 1 (1.9%) | 0.292 |
| Level of Education[3] | | | 0.075 |
| Low | 0 (0%)[N = 49] | 5 (9.8%)[N = 51] | |
| Middle | 22 (44.9%)[N = 49] | 25 (49%)[N = 51] | |
| High | 27 (55.1%)[N = 49] | 21 (41.2%)[N = 51] | |
| **IPAQ[4]** | | | 0.303 |
| Low physical activity | 3 (7%)[N = 43] | 11 (23.9%)[N = 46] | |
| Moderate physical activity | 19 (44.2%)[N = 43] | 14 (30.4%)[N = 46] | |
| High physical activity | 21 (48.8%)[N = 43] | 21 (45.7%)[N = 46] | |
| **BIA** | | | |
| BIA fm[(%)] | 30.7 (25.6–38) | 35.9 (29.05–40.2) | **0.026** |

(*Continued*)

**Table 1.** (Continued)

| | Women with uneventful pregnancies (N = 51) | Women with former PE/FGR (N = 53) | p** |
|---|---|---|---|
| BIA bcm(%) | 37.8 (33.8–40) | 34.7 (31.95–39.95) | 0.096 |
| **CIMT** | | | |
| CIMT average(mm) | 0.5 (0.5–0.6) | 0.55 (0.5–0.65)[N = 52] | 0.441 |
| **Basic serum analysis** | | | |
| HDL(mmol/l) | 1.60 (1.3–2.02) | 1.55 (1.16–1.79) | 0.228 |
| LDL(mmol/l) | 2.93 (2.42–3.52) | 2.96 (2.47–3.72) | 0.964 |
| LDL/HDL | 1.8 (1.4–2.4) | 1.9 (1.4–2.7) | 0.369 |
| Cholesterol(mmol/l) | 4.97 (4.31–5.8) | 4.86 (4.22–5.72) | 0.723 |
| GFR(ml/min) | 91.2 (82.9–101) | 93.5 (87–104.5) | 0.234 |
| Creatinine(mg/dl) | 69 (64–75) | 68 (61.5–72.5) | 0.298 |
| ASAT(U/l) | 0.35 (0.31–0.4) | 0.34 (0.3–0.39) | 0.476 |
| ALAT(U/l) | 0.3 (0.26–0.35) | 0.31 (0.23–0.42) | 0.997 |
| hs-CRP(mg/dl) | 1.15 (0.4–1.9) | 1.1 (0.5–2.7) | 0.441 |

*Data are n (%) or median (25th-75th percentile). Number of subjects (N) is given if deviating from indicated group size. Significant results by Mann-Whitney-U-Test ($p < 0.05$) are highlighted in bold.

**comparing uneventful pregnancies vs. pregnancies complicated by PE/FGR; [1]other medication including hormonal contraceptives, thyroid hormones, antidiabetics, blood thinners, diuretics, psychotropic drugs, antiepileptics, glucocorticoid sprays, antihistamines, NSAIDs, PPI, dietary supplements and herbal preparations; [2]arterial hypertension diagnosed and/or blood pressure $\geq$ 140/90 mmHg at study visit; [3]level of education low, middle school education; level of education middle, high school education; level of education high, general qualification for university entrance; [4] IPAQ high physical activity, vigorous intensity activity on at least 3 days achieving a minimum total physical activity of at least 1500 MET minutes a week or 7 or more days of any combination of walking, moderate intensity or vigorous intensity activities achieving a minimum total physical activity of at least 3000 MET minutes a week; IPAQ moderate physical activity, 3 or more days of vigorous intensity activity and/or walking of at least 30 minutes per day or 5 or more days of moderate intensity activity and/or walking of at least 30 minutes per day 5 or more days of any combination of walking, moderate intensity or vigorous intensity activities achieving a minimum total physical activity of at least 600 MET minutes a week; IPAQ low physical activity, if not meeting any of criteria for either moderate of high levels of physical activity; PE, preeclampsia; FGR, fetal growth restriction; PETN, pentaerythritol tetranitrate; BP, blood pressure; BMI, body mass index; IPAQ, International Physical Activity Questionnaire; BIA, bioelectrical impendance analysis; BIA fm, bioelectrical impendance analysis fat mass; BIA bcm, bioelectrical impendance analysis body cell mass; CIMT, carotid intima-media thickness; HDL, high-density lipoproteins; LDL, low-density lipoproteins; GFR, glomerular filtration rate; ASAT, aspartate-aminotransferase; ALAT, alanine-aminotransferase; hs-CRP, high-sensitive CRP

TTE results of women after PE/FGR with and without PETN use were each compared with the subgroup of women after uneventful pregnancies. (Table 4, Fig 2). TTE of affected women without previous medication showed a smaller mean left ventricular end-systolic diameter (LVESD, p = 0.021) and a larger mean left ventricular posterior wall diameter (LVPWD, p = 0.013) than those of the control group (Table 4, Fig 2). Comparing affected women with a history of PETN use to controls, TTE revealed smaller mean values for right atrial area (RA area, p = 0.006), right ventricular end-systolic diameter (RVESD, p = 0.048) and right ventricular end-diastolic diameter (RVEDD, p = 0.015). Results of USCOM and VICORDER analysis are displayed in S1 Table.

## Discussion

In this study, we examined the cardiovascular health of women 10–20 years after pregnancies complicated by PE/FGR in comparison to women following uneventful pregnancies. Although we found a significantly higher incidence of chronic diseases and arterial hypertension in women after PE/FGR, performed analyses of cardiovascular function did not show any significant differences between the groups. Considering the effect of PETN, we observed lower mean values for right atrial area and right ventricular diameters in affected women with former

**Table 2. Results of cardiovascular function analysis\*.**

| | Women with uneventful pregnancies (N = 51) | Women with former PE/FGR (N = 53) | p** |
|---|---|---|---|
| **Transthoracic echocardiography** | | | |
| HR(bpm) | 69 (61.5–76)N = 41 | 67 (63.25–75.75)N = 44 | 0.909 |
| CO(l/min) | 4.13 (2.6–4.76)N = 40 | 3.55 (2.83–4.53)N = 45 | 0.816 |
| SV(ml) | 56 (39.5–66.5)N = 41 | 56 (40–66)N = 45 | 0.883 |
| LVEDD M(mm) | 45.5 (42–48.25)N = 42 | 44.5 (40–48)N = 46 | 0.193 |
| LVEDD 4CH(mm) | 44 (41–47)N = 42 | 44 (42–46.25)N = 46 | 0.963 |
| LVESD M(mm) | 28 (26–30.5)N = 41 | 27 (24–29)N = 46 | 0.097 |
| LVESD 4CH(mm) | 29 (25.75–32)N = 42 | 28 (27–31)N = 46 | 0.700 |
| LVPWD M(mm) | 8 (8–9)N = 42 | 9 (8–10)N = 46 | **0.036** |
| LA area(cm2) | 16 (14–18)N = 42 | 17 (14–18)N = 46 | 0.576 |
| LA volume(ml) | 40.5 (33.5–50.5)N = 42 | 43 (30–48.25)N = 46 | 0.506 |
| RVEDD(mm) | 35 (31–38.5)N = 41 | 34 (31–36)N = 46 | 0.169 |
| RVESD(mm) | 22 (20.5–26)N = 41 | 22.5 (20–25.25)N = 46 | 0.505 |
| RA area(cm2) | 14 (12–15)N = 42 | 13 (11–14.25)N = 46 | 0.126 |
| lVSD(mm) | 10 (8–10)N = 42 | 10 (9–11)N = 46 | 0.062 |
| lVSD 4CH(mm) | 10 (8–10)N = 43 | 10 (9–11)N = 46 | 0.325 |
| LVEF 2CH(%) | 66 (62.75–69)N = 38 | 64 (60–67)N = 43 | 0.169 |
| LVEF 4CH(%) | 63.5 (60–67.25)N = 42 | 65 (59–67)N = 45 | 0.898 |
| TAPSE(mm) | 24 (21–25.25)N = 38 | 23 (20–25)N = 39 | 0.445 |
| sPAP(mmHg) | 17 (12.5–21.5)N = 29 | 19 (15–23)N = 31 | 0.324 |
| RVOT(mm) | 27 (25–28)N = 36 | 28 (25–30)N = 44 | 0.481 |
| PH(mm) | 20 (17.5–21)N = 21 | 20 (18–22.25)N = 30 | 0.537 |
| IVC(mm) | 13 (11–15.75)N = 36 | 13 (11–14.5)N = 45 | 0.958 |
| PVR(WU) | 1.06 (0.8–1.42)N = 29 | 1.18 (0.79–1.58)N = 32 | 0.461 |
| TRV(m/s) | 2.09 (1.78–2.33)N = 30 | 2.20 (1.94–2.4)N = 31 | 0.217 |
| TVI RVOT(cm) | 17 (15–19)N = 34 | 17 (14–20.25)N = 38 | 0.950 |
| E/A | 1.4 (1–1.79)N = 41 | 1.52 (1.25–1.7)N = 45 | 0.372 |
| DCT(ms) | 225 (178.5–250.5)N = 41 | 230 (194.5–262.5)N = 45 | 0.196 |
| E/ med E | 8.5 (6.48–10.15)N = 38 | 9.2 (7.6–11.35)N = 41 | 0.257 |
| E/ lat E | 6.32 (4.75–7.8)N = 30 | 6.5 (5.2–7.57)N = 34 | 0.554 |
| Dia. Dysfunction | 11 (28.2%)N = 39 | 9 (22.5%)N = 40 | 0.562 |
| **USCOM** | | | |
| HR(bpm) | 67 (62–72) | 66 (60–75) | 0.968 |
| SV(ml) | 74 (61–82) | 69 (68–77) | 0.110 |
| SVI(ml/m2) | 37 (33–44) | 36 (31–41) | 0.109 |
| CO(l/min) | 4.7 (4.2–5.6) | 4.4 (3.7–5.3) | 0.126 |
| CI(l/min/m2) | 2.6 (2.2–2.9) | 2.3 (2–2.8) | 0.056 |
| SVR(ds/cm5) | 1,787 (1469–2046) | 1,904 (1505–2398) | 0.145 |
| SVRI(ds/cm5*m2) | 3,267 (2711–6020) | 5,104(2845–7100) | 0.087 |
| VPK (m/s) | 1.1 (0.95–1.27) | 1.05 (0.93–1.27) | 0.388 |
| VTI (cm) | 25 (21–27) | 24 (20–26) | 0.388 |
| MD (m/min) | 16.3 (14–18.7) | 15 (12.3–18) | 0.221 |
| ET(%) | 39 (37–42) | 39 (35–42) | 0.827 |
| FTc(ms) | 376 (360–388) | 368 (347–389) | 0.827 |
| SVV(%) | 23 (17–28) | 24 (19–34) | 0.377 |
| SMII(W/m2) | 1.5 (1.3–1.8) | 1.4 (1.2–1.6) | 0.200 |
| **VICORDER** | | | |

*(Continued)*

**Table 2.** (Continued)

| | Women with uneventful pregnancies (N = 51) | Women with former PE/FGR (N = 53) | p** |
|---|---|---|---|
| PWV(m/s) | 5 (4–8)N = 49 | 06 (4–8.5) | 0.614 |
| Aix | 24 (19–31) | 24 (19.5–29) | 0.904 |
| AoPP(mmHg) | 64 (54–71) | 64 (59.5–72) | 0.522 |
| AoBP sys(mmHg) | 131 (119–144) | 137 (124–148.5) | 0.155 |
| AoBP dia(mmHg) | 68 (63–73) | 71 (62.5–78) | 0.171 |
| MAP(mmHg) | 95 (89–104) | 100 (92–109.5) | 0.089 |
| SV(ml) | 109 (97–127) | 113 (99–127) | 0.790 |
| CO(l/min) | 7 (6–8) | 7 (7–8) | 0.216 |
| CI(l/min/m2) | 4 (3–4) | 4 (3.5–5) | 0.217 |
| SEVR(%) | 158 (138–174) | 115 (109–130.5) | 0.130 |
| TPR(PRU) | 0.81 (0.69–0.99)N = 49 | 0.83 (0.73–0.95) | 0.574 |
| FMS(%) | 16 (9.25–22.75)N = 48 | 14 (9–21) | 0.364 |

*Data are n (%) or median (25th-75th percentile). Number of subjects (N) is given if deviating from indicated group size. Significant results by Mann-Whitney-U-Test ($p < 0.05$) are highlighted in bold

.**comparing uneventful pregnancies vs. pregnancies complicated by PE/FGR; PE, preeclampsia; FGR, fetal growth restriction; PETN, pentaerythritol tetranitrate; USCOM, Ultrasonic Cardiac Output Monitors; HR, heart rate; SV, stroke volume; SVI, stroke volume index; CO, cardiac output; CI, cardiac index; SVR, systemic vascular resistance; SVRI, systemic vascular resistance index; VPK, peak velocity of ventricular ejection; VTI, velocity time integral; MD, minute distance; ET, ejection time; $FT_c$, flow time corrected; SVV, stroke volume variation; SMII, Smith Madigan Inotropy Index; AoPP, aortic pulse pressure; AoBP sys, aortic blood pressure systolic; AoBP dia, aortic blood pressure diastolic; MAP, mean arterial pressure; Aix, augmentation index; SEVR, subendocardial viability ratio; TPR, total peripheral resistance; FMS, flow mediated slowing; PWV, pulse wave velocity; M, M-mode; 4CH, 4-chamber view; 2CH, 2-chamber view; LVEDD, left ventricular end-diastolic diameter; LVESD, left ventricular end-systolic diameter; LVPWD, left ventricular rear wall diameter; LA area, left atrial area; LA volume, left atrial volume; RVEDD, right ventricular end-diastolic diameter; RVESD, right ventricular end-systolic diameter; RA area, right atrial area; lVSD, interventricular septum diameter; LVEF, left ventricular ejection fraction; TAPSE, tricuspid annular plane systolic excursion; sPAP, systolic pulmonary arterial pressure; RVOT, right ventricular outflow tract; PH, pulmonary trunk diameter; IVC, inferior vena cava diameter; PVR, pulmonary vascular resistance; TRV, tricuspid regurgitant velocity; TVI RVOT, time-velocity integral of right ventricular outflow tract; E/A, E/A ratio; DCT, deceleration time; E/ med e´, E-wave velocity/ medial e´-velocity- ratio; E/ lat e´, E-wave velocity/ lateral e´-velocity-ratio; Dia. Disfunction, existence of diastolic dysfunction

PETN use compared to both, affected women without medication and healthy controls. However, all results were within reference ranges and should therefore be interpreted with caution.

Chronic arterial hypertension may reoccur or persist after the end of pregnancies complicated by PE [19]. Chronic hypertension is established as one of the major risk factors for the development of CVD [20]. Women after PE/FGR in our study population were not only significantly more likely to have arterial hypertension, but also to be hypertensive at time of study evaluation, regardless of established diagnosis or medication. In agreement with our data, a study focusing on cardiovascular risk after hypertensive disorders of pregnancy, including gestational hypertension and preeclampsia, found that risk of hypertension was 2.4 times higher in affected women 10 years after pregnancy compared to controls [21]. Similar findings were observed in women with a history of pregnancies complicated by FGR [22].

Contrary to our expectations, cardiovascular parameters of women after PE/FGR did not significantly differ from those of women after uneventful pregnancies. All results indicated an overall healthy cardiovascular status of our study population. Countouris et al. described an increased LVPWD in TTE of women after hypertensive pregnancy disorders almost one decade later [23]. Bokslag et al. confirmed this finding for a follow-up period of 9–16 years after delivery and additionally observed a development of diastolic dysfunction, detected as lower e´ and increased E/e' ratio [24], which we cannot confirm based on our results. However, it should be noted that mean follow-up age of cited studies differs slightly from that of

**Table 3. Group characteristics\*.**

| | Women with former PE/FGR without PETN (N = 40) | Women with former PE/FGR with PETN (N = 13) | p** |
|---|---|---|---|
| **Group Characteristics related to index pregnancy** | | | |
| Maternal Age at delivery[(years)] | 31 (27–34) | 32 (25–34.5) | 0.820 |
| Gestational age at delivery[(weeks of gestation)] | 34 (28–36) | 34 (29.5–37) | 0.648 |
| Birth Weight percentile child[(%)] | 7.5 (3–24.25) | 4 (2–13) | 0.165 |
| Presentation of pregnancy complication | | | |
| PE | 19 (47.5%) | 4 (30.8%) | |
| FGR | 8 (20%) | 5 (38.5%) | |
| PE and FGR | 13 (32.5%) | 4 (30.8%) | |
| Chronic diseases during index pregnancy | 26 (65%) | 6 (46.2%) | 0.232 |
| Diabetes mellitus | 1 (2.5%) | 0 (0%) | 0.569 |
| Arterial hypertension | 23 (57.5%) | 6 (46.2%) | 0.479 |
| Thyroid Diseases | 1 (2.5%) | 0 (0%) | 0.569 |
| Other Diseases | 4 (10%) | 1 (7.69%) | 0.806 |
| Medication intake during index pregnancy | 33 (82.5%) | 13 (100%) | 0.109 |
| PETN | 0 (0%) | 13 (100%) | **<0.001** |
| Antihypertensive medication | 9 (22.5%) | 3 (23.1%) | 0.966 |
| Aspirin | 3 (7.5%) | 2 (15.4%) | 0.403 |
| Other medication[1] | 22 (55%) | 0 (0%) | **<0.001** |
| **Characteristics at evaluation** | | | |
| Time since index pregnancy[(years)] | 14 (12–16) | 13 (8.5–14) | 0.072 |
| Age[(years)] | 44.5 (41–47.75) | 41 (38–47) | 0.133 |
| BMI[(kg/m2)] | 27.44 (23.16–33.01) | 24.84 (21.71–33.39) | 0.482 |
| BP systolic[(mmHg)] | 139 (123–153)[N = 39] | 136 (128.75–142.75) | 0.719 |
| BP diastolic[(mmHg)] | 89.50 (81.5–97)[N = 39] | 89.5 (83–96.25) | 0.949 |
| Hypertension[2] | 28 (71.8%)[N = 39] | 7 (53.8%) | 0.237 |
| Chronic diseases | 28 (70%) | 8 (61.5%) | 0.574 |
| Diabetes mellitus | 2 (5%) | 0 (0%) | 0.416 |
| Arterial hypertension | 17 (42.5%) | 5 (38.5%) | 0.799 |
| Thyroid Diseases | 11 (27.5%) | 1 (7.7%) | 0.142 |
| Lipid Metabolism Diseases | 1 (2.5%) | 0 (0%) | 0.569 |
| On Medication | 31 (77.5%) | 6 (46.2%) | **0.034** |
| Antihypertensive medication | 18 (45%) | 4 (30.8%) | 0.370 |
| Other medication[1] | 23 (57.5%) | 5 (38.5%) | 0.370 |
| Consumption behaviour | | | |
| Smoking | 8 (20%) | 2 (15.4%) | 0.579 |
| Alcohol consumption | 28 (70%) | 10 (76.9%) | 0.633 |
| Drug consumption | 0 (0%) | 1 (07.7%) | 0.079 |
| Level of Education[3] | | | 0.402 |
| Low | 3 (7.7%)[N = 39] | 2 (16.7%)[N = 12] | |
| Middle | 19 (48.7%)[N = 39] | 6 (50%)[N = 12] | |
| High | 17 (43.6%)[N = 39] | 4 (33.3%)[N = 12] | |
| **IPAQ[4]** | | | 0.859 |
| Low physical activity | 9 (25.7%)[N = 35] | 2 (18.2%)[N = 11] | |
| Moderate physical activity | 10 (28.6%)[N = 35] | 4 (36.4%)[N = 11] | |
| High physical activity | 16 (45.7%)[N = 35] | 5 (45.5%)[N = 11] | |
| **BIA** | | | |
| BIA fm[(%)] | 36.15 (30.25–40.15) | 34.6 (26.45–40.3) | 0.699 |

(*Continued*)

**Table 3.** (Continued)

| | Women with former PE/FGR without PETN (N = 40) | Women with former PE/FGR with PETN (N = 13) | p** |
|---|---|---|---|
| BIA bcm (%) | 34.5 (32.53–39.43) | 35.4 (31.7–41.3) | 0.732 |
| **CIMT** | | | |
| CIMT average (mm) | 0.55 (0.5–0.65) N = 39 | 0.5 (0.45–0.6) | 0.128 |
| **Basic serum analysis** | | | |
| HDL (mmol/l) | 1.56 (1.25–1.79) | 1.55 (1.34–1.81) | 0.657 |
| LDL (mmol/l) | 3.02 (2.7–3.76) | 2.48 (1.73–3.2) | **0.028** |
| LDL/HDL | 2.25 (1.7–2.8) | 1.5 (1.25–2.25) | 0.056 |
| Cholesterol (mmol/l) | 4.99 (4.39–5.83) | 4.31 (3.48–5.07) | **0.044** |
| GFR (ml/min) | 92.95 (87.15–104.33) | 97.1 (84.9–110.8) | 0.605 |
| Creatinine (mg/dl) | 68 (62.25–72.75) | 67 (58.5–73) | 0.725 |
| ASAT (U/l) | 0.34 (0.31–0.39) | 0.33 (0.29–0.38) | 0.341 |
| ALAT (U/l) | 0.32 (0.26–0.42) | 0.25 (0.21–0.39) | 0.203 |
| hs-CRP (mg/dl) | 1.2 (0.43–2.98) | 1.1 (0.55–2.7) | 0.901 |

*Data are n (%) or median (25th-75th percentile). Number of subjects (N) is given if deviating from indicated group size. Significant results by Mann-Whitney-U-Test ($p < 0.05$) are highlighted in bold.

**comparing pregnancies complicated by PE/FGR without PETN intake vs. pregnancies complicated by PE/FGR with PETN intake

[1] other medication including hormonal contraceptives, thyroid hormones, antidiabetics, blood thinners, diuretics, psychotropic drugs, antiepileptics, glucocorticoid sprays, antihistamines, NSAIDs, PPI, dietary supplements and herbal preparations

[2] arterial hypertension diagnosed and/or blood pressure $\geq$ 140/90 mmHg at study visit

[3] level of education low, middle school education; level of education middle, high school education; level of education high, general qualification for university entrance

[4] IPAQ high physical activity, vigorous intensity activity on at least 3 days achieving a minimum total physical activity of at least 1500 MET minutes a week or 7 or more days of any combination of walking, moderate intensity or vigorous intensity activities achieving a minimum total physical activity of at least 3000 MET minutes a week; IPAQ moderate physical activity, 3 or more days of vigorous intensity activity and/or walking of at least 30 minutes per day or 5 or more days of moderate intensity activity and/or walking of at least 30 minutes per day 5 or more days of any combination of walking, moderate intensity or vigorous intensity activities achieving a minimum total physical activity of at least 600 MET minutes a week; IPAQ low physical activity, if not meeting any of criteria for either moderate of high levels of physical activity; PE, preeclampsia; FGR, fetal growth restriction; PETN, pentaerythritol tetranitrate; BP, blood pressure; BMI, body mass index; IPAQ, International Physical Activity Questionnaire; BIA, bioelectrical impendance analysis; BIA fm, bioelectrical impendance analysis fat mass; BIA bcm, bioelectrical impendance analysis body cell mass; CIMT, carotid intima-media thickness; HDL, high-density lipoproteins; LDL, low-density lipoproteins; GFR, glomerular filtration rate; ASAT, aspartate-aminotransferase; ALAT, alanine-aminotransferase; hs-CRP, high-sensitive CRP

ours, which could have an influence on differing results if one assumes that the long-term effect of PE/FGR also correlates with the age of onset. Other studies have shown an association between a history of PE and concentric left ventricular remodeling as well as preclinical heart failure 4 to 10 years after delivery, however, these associations were attenuated after adjustment for age, BMI, and current hypertension [25]. Similarly, our studies on arterial function, which is known to be an important marker in the assessment of CVD risk, showed no differences in commonly used indices of large artery elasticity and regulation of vascular tone, via arterial stiffness, arterial wave reflection and endothelial function testing [26]. In agreement to our observations, Östlund et al. performed pulse wave analysis and endothelial assessment by flow-mediated dilation on the forearm in 15 women after PE and in 16 matched controls both 1 year and 11 years after delivery. They found that vascular stiffness and endothelial impairment observed in the PE-group after one year, normalized at year 11 [27]. A study by Christensen et al. 10 years post-PE, could not find any abnormalities in markers of arterial stiffness via aortic PWV and Aix or of atherosclerosis measures via CIMT and carotid plaque presence in women after PE compared to controls [28]. Speculating why cardiovascular changes were not observed in our population we conclude that the follow-up period might

**Table 4. Results of cardiovascular function analysis*.**

| | Women with uneventful pregnancies (N = 51) | Women with former PE/FGR without PETN (N = 40) | p** | Women with former PE/FGR with PETN (N = 13) | p*** |
|---|---|---|---|---|---|
| **Transthoracic echocardiography** | | | | | |
| HR (bpm) | 69 (61.5–76) N = 41 | 66.5 (63–75.8) N = 32 | 0.911 | 70.5 (64.8–77) N = 12 | 0.625 |
| CO (l/min) | 4.1 (2.6–4.8) N = 40 | 3.4 (2.7–4.5) N = 33 | 0.492 | 4.4 (3.4–4.5) N = 12 | 0.441 |
| SV (ml) | 56 (39.5–66.5) N = 41 | 53 (36–66.5) N = 33 | 0.900 | 57 (50.8–64.8) N = 12 | 0.545 |
| LVEDD M (mm) | 45.5 (42–48.3) N = 42 | 45 (40–47.3) N = 34 | 0.167 | 44 (40.5–48) N = 12 | 0.624 |
| LVEDD 4CH (mm) | 44 (41–47) N = 42 | 45 (42.8–47.3) N = 34 | 0.512 | 43 (37–45.5) N = 12 | 0.234 |
| LVESD M (mm) | 28 (26–30.5) N = 41 | 26 (23–28.3) N = 34 | **0.021** | 28.5 (27–29.8) N = 12 | 0.646 |
| LVESD 4CH (mm) | 29 (25.8–32) N = 42 | 28 (27–31) N = 34 | 0.773 | 27.5 (27–32) N = 12 | 0.669 |
| LVPWD M (mm) | 8 (8–9) N = 42 | 9 (8–10) N = 34 | **0.013** | 8 (8–9) N = 12 | 0.766 |
| LA area (cm2) | 16 (14–18) N = 42 | 17.5 (14–19) N = 34 | 0.205 | 14.5 (13–17.5) N = 12 | 0.258 |
| LA volume (ml) | 40.5 (33.5–50.5) N = 42 | 45.5 (31.5–49.3) N = 34 | 0.867 | 36.5 (29.3–46.3) N = 12 | 0.186 |
| RVEDD (mm) | 35 (31–38.5) N = 41 | 35 (31.8–37) N = 34 | 0.611 | 32 (29.3–33.8) N = 12 | **0.015** |
| RVESD (mm) | 22 (20.5–26) N = 41 | 24 (20–26) N = 34 | 0.877 | 20.5 (19.3–22.5) N = 12 | **0.048** |
| RA area (cm2) | 14 (12–15) N = 42 | 13 (11–15) N = 34 | 0.595 | 11 (10–12.8) N = 12 | **0.006** |
| lVSD (mm) | 10 (8–10) N = 42 | 10 (9–11) N = 34 | 0.064 | 10 (9–11) N = 12 | 0.333 |
| lVSD 4CH (mm) | 10 (8–10) N = 43 | 10 (9–11) N = 34 | 0.278 | 9 (8.3–10.3) N = 12 | 0.771 |
| LVEF 2CH (%) | 66 (62.8–69) N = 38 | 64.5 (60–67) N = 32 | 0.306 | 62 (60–67) N = 11 | 0.159 |
| LVEF 4CH (%) | 63.5 (60–67.3) N = 42 | 65 (59–67.5) N = 33 | 0.802 | 63 (58.8–66) N = 12 | 0.859 |
| TAPSE (mm) | 24 (21–25.3) N = 38 | 23 (21–26.3) N = 30 | 0.857 | 20 (20–24) N = 9 | 0.109 |
| sPAP (mmHg) | 17 (12.5–21.5) N = 29 | 19 (16–23) N = 25 | 0.213 | 16 (12.5–20.8) N = 6 | 0.848 |
| RVOT (mm) | 27 (25–28) N = 36 | 28 (26–30.5) N = 33 | 0.150 | 25 (24–29) N = 11 | 0.247 |
| PH (mm) | 20 (17.5–21) N = 21 | 21 (18.5–23) N = 21 | 0.229 | 19 (18–20.5) N = 9 | 0.504 |
| IVC (mm) | 13 (11–15.8) N = 36 | 13 (11–16) N = 33 | 0.861 | 13 (10.3–14) N = 12 | 0.631 |
| PVR (WU) | 1.1 (0.8–1.4) N = 29 | 1.2 (0.6–1.5) N = 27 | 0.628 | 1.4 (1–1.7) N = 5 | 0.318 |
| TRV (m/s) | 2.1 (1.8–2.3) N = 30 | 2.2 (2–2.4) N = 25 | 0.119 | 2 (1.8–2.3) N = 6 | 0.788 |
| TVI RVOT (cm) | 17 (15–19) N = 34 | 17 (14–21.3) N = 30 | 0.845 | 17 (14.5–18) N = 8 | 0.789 |
| E/A | 1.4 (1–1.8) N = 41 | 1.5 (1.3–1.7) N = 33 | 0.244 | 1.5 (1–1.7) N = 12 | 0.932 |
| DCT (ms) | 225 (179–251) N = 41 | 239 (194.5–265.5) N = 33 | 0.169 | 227 (185.3–256.3) N = 12 | 0.625 |
| E/ med E | 8.5 (6.5–10.2) N = 38 | 9.3 (7.2–11.5) N = 30 | 0.224 | 8.6 (7.8–10.3) N = 11 | 0.684 |
| E/ lat E | 6.3 (4.8–7.8) N = 30 | 6.6 (5.3–8) N = 27 | 0.502 | 6.4 (4.9–7.4) N = 7 | 0.955 |
| Dia. Dysfunction | 11 (28.2%) N = 39 | 7 (24.1%) N = 29 | 0.709 | 2 (18.2%) N = 11 | 0.508 |

*Data are n (%) or median (25th-75th percentile). Number of subjects (N) is given if deviating from indicated group size. Significant results by Mann-Whitney-U-Test ($p < 0.05$) are highlighted in bold. P

**comparing uneventful pregnancies vs. pregnancies complicated by PE/FGR without PETN intake; p

***comparing uneventful pregnancies vs. pregnancies complicated by PE/FGR with PETN intake; PE, preeclampsia; FGR, fetal growth restriction; PETN, pentaerythritol tetranitrate; TTE, transthoracic echocardiography; HR, heart rate; CO, cardiac output; SV, stroke volume; M, M-mode; 4CH, 4-chamber view; 2CH, 2-chamber view; LVEDD, left ventricular end-diastolic diameter; LVESD, left ventricular end-systolic diameter; LVPWD, left ventricular rear wall diameter; LA area, left atrial area; LA volume, left atrial volume; RVEDD, right ventricular end-diastolic diameter; RVESD, right ventricular end-systolic diameter; RA area, right atrial area; lVSD, interventricular septum diameter; LVEF, left ventricular ejection fraction; TAPSE, tricuspid annular plane systolic excursion; sPAP, systolic pulmonary arterial pressure; RVOT, right ventricular outflow tract; PH, pulmonary trunk diameter; IVC, inferior vena cava diameter; PVR, pulmonary vascular resistance; TRV, tricuspid regurgitant velocity; TVI RVOT, time-velocity integral of right ventricular outflow tract; E/A, E/A ratio; DCT, deceleration time; E/ med e´, E-wave velocity/ medial e´-velocity- ratio; E/ lat e´, E-wave velocity/ lateral e´-velocity-ratio; Dia. Disfunction, existence of diastolic dysfunction

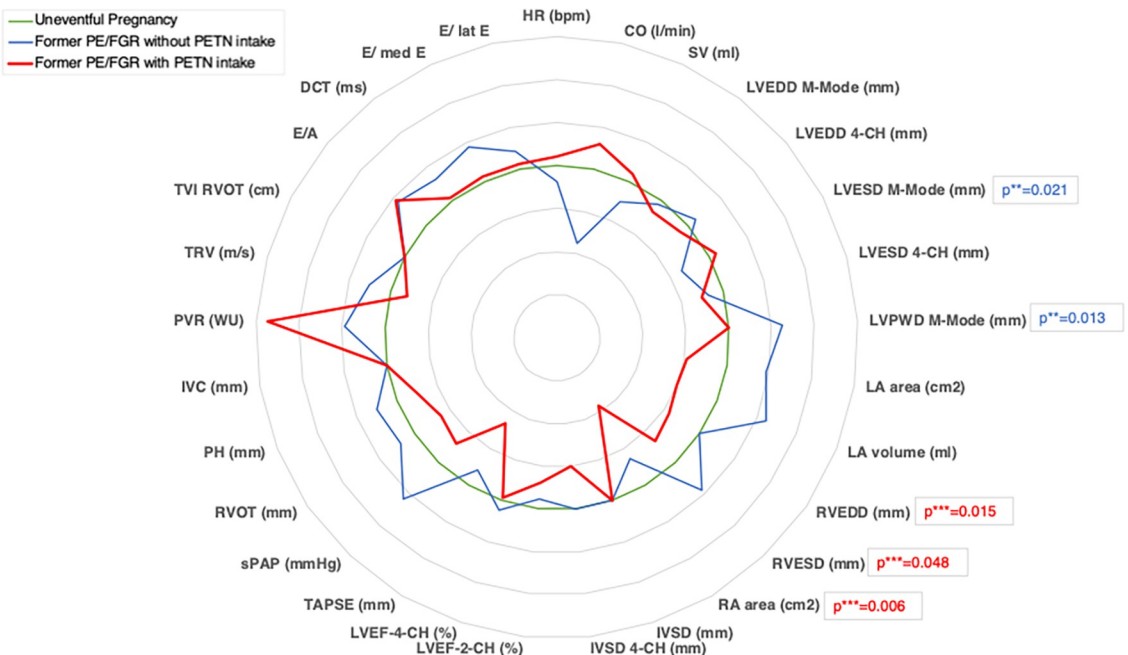

**Fig 2. Radar chart visually demonstrating the position of mean values of TTE parameters in relation to normalized outcomes of women with uneventful pregnancies.** Green chart, women with uneventful pregnancies; blue chart, women with PE/FGR without PETN intake in pregnancy; red chart, women with PE/FGR with PETN intake in pregnancy; Only significant results of Mann-Whitney U test (p < 0.05) are accentuated. **comparing uneventful pregnancies vs. pregnancies with PE/FGR without PETN intake; ***comparing uneventful pregnancies vs. pregnancies with PE/FGR with PETN intake; TTE, transthoracic echocardiography; PE, preeclampsia; FGR, fetal growth restriction; PETN, pentaerythritol tetranitrate; HR, heart rate; CO, cardiac output; SV, stroke volume; 4-CH, 4-chamber view; 2-CH, 2-chamber view; LVEDD, left ventricular end-diastolic diameter; LVESD, left ventricular end-systolic diameter; LVPWD, left ventricular rear wall diameter; LA area, left atrial area; LA volume, left atrial volume; RVEDD, right ventricular end-diastolic diameter; RVESD, right ventricular end-systolic diameter; RA area, right atrial area; lVSD, interventricular septum diameter; LVEF, left ventricular ejection fraction; TAPSE, tricuspid annular plane systolic excursion; sPAP, systolic pulmonary arterial pressure; RVOT, right ventricular outflow tract; PH, pulmonary trunk diameter; IVC, inferior vena cava diameter; PVR, pulmonary vascular resistance; TRV, tricuspid regurgitant velocity; TVI RVOT, time-velocity integral of right ventricular outflow tract; E/A, E/A ratio; DCT, deceleration time; E/ med e, E-wave velocity/ medial e-velocity- ratio; E/ lat e, E-wave velocity/ lateral e-velocity-ratio.

have been too long to detect short-term postpartum changes, as studies have shown recovery of cardiac abnormalities caused by gestational hypertension [29]. Simultaneously, the selected time period might have been too short to observe long-term effects of predisposing risk factors, such as arterial hypertension on the cardiovascular system.

As the importance of cardiovascular follow-up after high-risk pregnancies gains growing awareness among obstetricians, the choice of appropriate assessment methods must be clarified. TTE is known as the most commonly used procedure to analyze cardiac structure and function and is therefore considered to be the gold standard method [30]. However, correct application and interpretation of results requires a great amount of experience, time, and is highly examiner-dependent. In order to explore new semi- to fully-automated measurement methods, providing results comparable to the gold standard method, we selected USCOM and VICORDER in addition to conventional TTE. Analyzing our results, different assessment methods gave different and partly contradictory results (e.g. Table 2), which is why we refer to measurements of TTE when discussing our results. Although several studies confirm the comparability of the methods, our study results do not fully support this. It must be noted, however, that all obtained values were within reference ranges. Inter-examiner variance and a

highly variable physiognomy of our study participants may have influenced feasibility and quality of measurement results, especially in USCOM analysis.

The organic nitrate pentaerythrityl tetranitrate (PETN) is widely used in the treatment of CVD and has been shown to improve cardiovascular function in humans and rats and to have protective effects on human endothelial cells [31]. It improves blood flow and oxygenation of the myocardium through vasodilator effects and also has endothelial protective properties by increasing the expression of antioxidant genes such as heme oxygenase-1 (HO-1) in human endothelial cells [32]. The Valensise group demonstrated an improvement in maternal haemodynamics in pregnancies complicated by gestational hypertension when NO donors were added to initiated antihypertensive therapy, reducing maternal total vascular resistance (TVR) and increasing maternal cardiac output (CO) 5–14 days after initiation of treatment [33]. When investigating a possible long-term effect of PETN in our collective, we found that women who had taken the medication during pregnancy showed smaller right atria and ventricle heart size, compared to women with former uneventful pregnancies (Table 4, Fig 2). Interestingly, these differences were also found in direct comparison between the PE/FGR groups with and without medication (S2 Table). Hypothetically, if these observations were a consequence of previous PETN use, a possible pathophysiological theory could be that drug-induced reduction in cardiac afterload, due to vasodilation and reduction of TVR, must have led to reduced cardiac stress during pregnancy, to which the more vulnerable right heart responded with a slighter enlargement than in women not taking PETN. Although these observations may be of phenomenological interest rather than biological relevance, this effect appears to persist for years after pregnancy and might still be relevant. In a multicentre trial investigating the effect of PETN in high-risk pregnancies, it was clearly demonstrated that not only the incidence of pregnancy-induced hypertension and PE was reduced in women taking PETN, also Kaplan-Meier analysis showed that blood pressure remained significantly longer within normal range when taking PETN [6]. Following the hypothesis that cardiac remodelling during pregnancy is altered by maternal blood pressure, possibly crossing the boundary between reversible myocardial remodelling to irreversible hypertrophy and between physiological enlargement of the heart cavities to non-physiological irreversible dilation, the lowering of blood pressure by PETN during pregnancy could serve as an explanation for our observation. A larger sample size is required to demonstrate these long-term effects of PETN on maternal heart structure and to better understand their significance.

## Strengths and limitations

The study design was characterized by an overall large cohort with a median follow-up period of 14 years postpartum. We provided extensive data on the cardiovascular health status within this cohort, applying numerous assessment methods. While there was an almost balanced group comparison of former healthy to diseased pregnant women, it must be critically considered that the PETN group was relatively small, making results less representative. Due to the small number of events, statistical analyses are rather exploratory, which also made it difficult to apply multiple testing correction. This is why we primarily used simple analysis methods with individual consideration of influencing variables. Furthermore, due to lack of data, it cannot be conclusively assessed how clinical profile and parameters of study participants differed before index pregnancies. The availability of pre-existing cardiovascular data would have allowed for a better differentiation between possible predispositions to pregnancy effects on the maternal organisms. In addition, the participants in the PETN pilot study suffered from less severe PE, which led to heterogeneity in the PE cohort. Analyzing the results of applied methods for cardiovascular function diagnostics, partly contradictory differences became

apparent. Due to this fact, further investigation on correlation and equivalence of methods in clinical application is imperative.

## Conclusions

In summary, we present data on cardiovascular follow-up of a well-defined cohort from a single center 10–20 years after pregnancies complicated by PE. Our data confirm that these women are at risk of lifelong arterial hypertension, which is a relevant risk for onset of CVD. These data emphasize the importance of comprehensive cardiovascular follow-up of women with a history of placenta associated pregnancy diseases. In this context, modern techniques such as USCOM and VICORDER analysis appear particularly attractive due to their ease of use, but comparability with conventional methods should be better investigated before they are established in routine care. Our data on the potential long-term effects of PETN use in pregnancy on women's cardiovascular anatomy support the concept that improved cardiovascular blood pressure control during pregnancy may influence long-term maternal outcomes.

## Supporting information

**S1 Checklist. TREND statement checklist.**
(PDF)

**S1 Table. Results of cardiovascular function analysis.** *Data are n (%) or median (25th-75th percentile). Number of subjects (N) is given if deviating from indicated group size. Significant results by Mann-Whitney-U-Test ($p < 0.05$) are highlighted in bold. p**comparing uneventful pregnancies vs. pregnancies complicated by PE/FGR without PETN intake; p***comparing uneventful pregnancies vs. pregnancies complicated by PE/FGR with PETN intake; PE, preeclampsia; FGR, fetal growth restriction; PETN, pentaerythritol tetranitrate; USCOM, Ultrasonic Cardiac Output Monitors; HR, heart rate; SV, stroke volume; SVI, stroke volume index; CO, cardiac output; CI, cardiac index; SVR, systemic vascular resistance; SVRI, systemic vascular resistance index; VPK, peak velocity of ventricular ejection; VTI, velocity time integral; MD, minute distance; ET, ejection time; FT$_c$, flow time corrected; SVV, stroke volume variation; SMII, Smith Madigan Inotropy Index; VICORDER; AoPP, aortic pulse pressure; AoBP sys, aortic blood pressure systolic; AoBP dia, aortic blood pressure diastolic; MAP, mean arterial pressure; Aix, augmentation index; SEVR, subendocardial viability ratio; TPR, total peripheral resistance; FMS, flow mediated slowing; PWV, pulse wave velocity.
(PDF)

**S2 Table. Results of cardiovascular function analysis.** *Data are n (%) or median (25th-75th percentile). Number of subject (N) is given if deviating from indicated group size. Significant results by Mann-Whitney-U-Test ($p < 0.05$) are highlighted in bold. p**comparing pregnancies complicated by PE/FGR without PETN intake vs. pregnancies complicated by PE/FGR with PETN intake; PE, preeclampsia; FGR, fetal growth restriction; PETN, pentaerythritol tetranitrate; USCOM, Ultrasonic Cardiac Output Monitors; HR, heart rate; SV, stroke volume; SVI, stroke volume index; CO, cardiac output; CI, cardiac index; SVR, systemic vascular resistance; SVRI, systemic vascular resistance index; VPK, peak velocity of ventricular ejection; VTI, velocity time integral; MD, minute distance; ET, ejection time; FT$_c$, flow time corrected; SVV, stroke volume variation; SMII, Smith Madigan Inotropy Index; AoPP, aortic pulse pressure; AoBP sys, aortic blood pressure systolic; AoBP dia, aortic blood pressure diastolic; MAP, mean arterial pressure; Aix, augmentation index; SEVR, subendocardial viability ratio; TPR, total peripheral resistance; FMS, flow mediated slowing; PWV, pulse wave velocity; M, M-mode; 4CH, 4-chamber view; 2CH, 2-chamber view; LVEDD, left ventricular end-diastolic

diameter; LVESD, left ventricular end-systolic diameter; LVPWD, left ventricular rear wall diameter; LA area, left atrial area; LA volume, left atrial volume; RVEDD, right ventricular end-diastolic diameter; RVESD, right ventricular end-systolic diameter; RA area, right atrial area; lVSD, interventricular septum diameter; LVEF, left ventricular ejection fraction; TAPSE, tricuspid annular plane systolic excursion; sPAP, systolic pulmonary arterial pressure; RVOT, right ventricular outflow tract; PH, pulmonary trunk diameter; IVC, inferior vena cava diameter; PVR, pulmonary vascular resistance; TRV, tricuspid regurgitant velocity; TVI RVOT, time-velocity integral of right ventricular outflow tract; E/A, E/A ratio; DCT, deceleration time; E/ med e´, E-wave velocity/ medial e´-velocity- ratio; E/ lat e´, E-wave velocity/ lateral e´-velocity-ratio; Dia. Disfunction, existence of diastolic dysfunction.
(PDF)

**S1 File.**
(PDF)

## Author Contributions

**Conceptualization:** Anna Multhaup, Thomas Lehmann, Ekkehard Schleußner, Marcus Franz, Tanja Groten.

**Data curation:** Charlotte Lößner, Anna Multhaup, Yvonne Heimann, Tanja Groten.

**Formal analysis:** Charlotte Lößner, Thomas Lehmann.

**Funding acquisition:** Anna Multhaup, Tanja Groten.

**Investigation:** Charlotte Lößner, Anna Multhaup, Laura Bäz, Tanja Groten.

**Methodology:** Thomas Lehmann, Marcus Franz.

**Project administration:** Marcus Franz.

**Resources:** Laura Bäz, Ekkehard Schleußner.

**Supervision:** Marcus Franz, Tanja Groten.

**Visualization:** Yvonne Heimann.

**Writing – original draft:** Charlotte Lößner, Anna Multhaup, Tanja Groten.

**Writing – review & editing:** Anna Multhaup, Laura Bäz, Thomas Lehmann, Ekkehard Schleußner, Marcus Franz, Tanja Groten.

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
