## [Decision Letter · Decision Letter 0]

5 Jul 2024

PONE-D-24-10406Cardiovascular health of women 10 to 20 years after placenta-related pregnancy diseases considering the possible effect of pentaerythrityl tetranitrate treatment during pregnancy on long-term maternal cardiovascular health (PAVA study)PLOS ONE

Dear Dr. Groten,

Thank you for submitting your manuscript to PLOS ONE. After careful consideration, we feel that it has merit but does not fully meet PLOS ONE’s publication criteria as it currently stands. Therefore, we invite you to submit a revised version of the manuscript that addresses the points raised during the review process.

We look forward to receiving your revised manuscript.

Kind regards,

Edina Cenko, M.D., Ph.D.

Academic Editor

PLOS ONE

“This work was funded by the Deutsche Forschungsgemeinschaft (DFG, German Research Foundation) Clinician Scientist Program OrganAge funding number 413668513 and by the Interdisciplinary Center of Clinical Research of the Medical Faculty Jena to AM.”

4. Please amend the manuscript submission data (via Edit Submission) to include authors Laura Bäz, MD, Thomas Lehmann, PHD, Yvonne Heimann, Ms., Ekkehard Schleußner, Prof. and Marcus Franz, Prof.

5. Please amend your authorship list in your manuscript file to include author Universitätsklinikum Jena.

Reviewers' comments:

Reviewer's Responses to Questions

**Comments to the Author**

1. Is the manuscript technically sound, and do the data support the conclusions?

Reviewer #1: Yes

Reviewer #2: Partly

2. Has the statistical analysis been performed appropriately and rigorously? 

Reviewer #1: Yes

Reviewer #2: No

3. Have the authors made all data underlying the findings in their manuscript fully available?

Reviewer #1: No

Reviewer #2: Yes

4. Is the manuscript presented in an intelligible fashion and written in standard English?

Reviewer #1: Yes

Reviewer #2: Yes

5. Review Comments to the Author

Reviewer #1: This single center study compared 53 participants after PE/FGR and 51 controls

That were followed up at an average of 14 years after index pregnancies. The results confirmed the previous finding that the risk of CVD is increased in women after PE/FGR compared to women after uneventful pregnancies. However, there were no differences in cardiovascular function observed, which is unexpected and contradictory to that of some previous studies. The authors also put a cautionary note on interpreting the observation that affected women with PETN intake during pregnancy showed lower mean values for right atrial area and ventricle in comparison to controls and also to affected women without former medication. Overall, it is a well carried out study with meticulous analysis.

I have some minor comments—

1) Multiple testing correction was not taken in this study. Note that the subtle changes observed for those with PETN would became insignificant if multiple testing correction procedure was taken.

2) In the current study the age at the follow-up evaluation is with a mean of ~44-45 and range of 40 to 49. In Courntouris et al. (2021), the mean age was ~48, and in Tokslage et al. (2017) it is above 50s’. Can the difference observed between the current study and the previously study due to the age difference? A hypothesis is that the long-term effect of PE/FGR is also correlated with age of onset.

Reviewer #2: In this manuscript, the authors present interesting data from the PAVA study, with particular focus on long-term outcomes following placenta-related complications during pregnancy. The authors also conducted a sub-analyisis investigating the potential role of pentaerythrityl tetranitrate in influencing prognosis.

This particular topic is of interest, as far too many women face potentially life-threatening complications during pregnancy, and too few data on potential treatment strategies for these conditions are currently available.

However, the results of this analysis are based on a very small sample, especially considering the number of pregnant women facing the diagnosis of pre-eclampsia each year. This becomes more evident after noticing that only crude statistical analyses could be conducted. Maybe, using these data as a companion to a meta-analysis of currently available data could allow for slightly stronger conclusions.

6. PLOS authors have the option to publish the peer review history of their article (what does this mean?). If published, this will include your full peer review and any attached files.

Reviewer #1: No

Reviewer #2: No

---

## [Author Response · Author response to Decision Letter 0]

4 Aug 2024

Dear Editor and Reviewers, 

we like to thank you for the valuable comments and are convinced that the manuscript is now further improved. Please find the detailed responses below. 

Best Regards Tanja Groten in behalf of the authors

1. Reviewers' comments:

Reviewer #1: This single center study compared 53 participants after PE/FGR and 51 controls

That were followed up at an average of 14 years after index pregnancies. The results confirmed the previous finding that the risk of CVD is increased in women after PE/FGR compared to women after uneventful pregnancies. However, there were no differences in cardiovascular function observed, which is unexpected and contradictory to that of some previous studies. The authors also put a cautionary note on interpreting the observation that affected women with PETN intake during pregnancy showed lower mean values for right atrial area and ventricle in comparison to controls and also to affected women without former medication. Overall, it is a well carried out study with meticulous analysis.

We thank the reviewer for this appreciative comment.

Comments:

1) Multiple testing correction was not taken in this study. Note that the subtle changes observed for those with PETN would became insignificant if multiple testing correction procedure was taken.

We thank the reviewer for that comment. Not using multiple testing correction is indeed a limitation. Due to the small number of events, the analyses should be seen as exploratory. This also makes it more difficult to apply multiple analysis methods, which is why we primarily used simple analysis methods. We have added a comment to this fact in the discussion (page 22, line 374-377).

2) In the current study the age at the follow-up evaluation is with a mean of ~44-45 and range of 40 to 49. In Courntouris et al. (2021), the mean age was ~48, and in Tokslage et al. (2017) it is above 50s’. Can the difference observed between the current study and the previously study due to the age difference? A hypothesis is that the long-term effect of PE/FGR is also correlated with age of onset.

We thank the reviewer for this thought. This could well be true, but of course it is not possible to answer this question with our data set. It could also be a fact, described by others, that the differences between groups are even greater 5 years after pregnancy and probably decrease at later time points, as we have pointed out in our discussion. However, we have added a sentence commenting on the age issue in the discussion (page 18, line 293-296).

Reviewer #2: In this manuscript, the authors present interesting data from the PAVA study, with particular focus on long-term outcomes following placenta-related complications during pregnancy. The authors also conducted a sub-analysis investigating the potential role of pentaerythrityl tetranitrate in influencing prognosis. This particular topic is of interest, as far too many women face potentially life-threatening complications during pregnancy, and too few data on potential treatment strategies for these conditions are currently available. However, the results of this analysis are based on a very small sample, especially considering the number of pregnant women facing the diagnosis of pre-eclampsia each year. This becomes more evident after noticing that only crude statistical analyses could be conducted. Maybe, using these data as a companion to a meta-analysis of currently available data could allow for slightly stronger conclusions.

We thank the reviewer for this valuable comment. We are aware that our cohort, particularly the sub-cohort taking PETN during their index pregnancy, is particularly small. For this reason, we were cautious in formulating our conclusions but were still able to point to some significant findings. However, we are aware that our data cannot be generalized at this point. Due to the small number of events, analyses are rather exploratory, which is why we primarily used simple analysis methods with individual consideration of influencing variables. We have added a comment on this limitation to the discussion (page 22, line 374-377). We are happy to take up the suggestion of conducting a meta-analysis for a future publication.

1. Please ensure that your manuscript complies with the PLOS ONE style guide, including the file naming guidelines.

Editing is done analogue to PLOS ONE style sheets.

2. Thank you for stating the following financial disclosure: “This work was funded by the Deutsche Forschungsgemeinschaft (DFG, German Research Foundation) Clinician Scientist Program OrganAge funding number 413668513 and by the Interdisciplinary Center of Clinical Research of the Medical Faculty Jena to AM.” Please state what role the funders took in the study. If the funders had no role, please state: "The funders had no role in study design, data collection and analysis, decision to publish, or preparation of the manuscript." If this statement is not correct you must amend it as needed. Please include this amended Role of Funder statement in your cover letter; we will change the online submission form on your behalf.

The requested financial information has been processed.

All authors agreed in advance on a data sharing plan in case of paper acceptance.

4. Please amend the manuscript submission data (via Edit Submission) to include authors Laura Bäz, MD, Thomas Lehmann, PHD, Yvonne Heimann, Ms., Ekkehard Schleußner, Prof. and Marcus Franz, Prof.

The manuscript submission dates have been changed according to requested requirements.

Appropriate information on supplemental material has been included within the manuscript.

The reference list was again verified for completeness and accuracy.

---

## [Editor Report · Decision Letter 1]

7 Aug 2024

Cardiovascular health of women 10 to 20 years after placenta-related pregnancy diseases considering the possible effect of pentaerythrityl tetranitrate treatment during pregnancy on long-term maternal cardiovascular health (PAVA study)

PONE-D-24-10406R1

Dear Dr. Groten,

We’re pleased to inform you that your manuscript has been judged scientifically suitable for publication and will be formally accepted for publication once it meets all outstanding technical requirements.

Kind regards,

Edina Cenko, M.D., Ph.D.

Academic Editor

PLOS ONE

---

## [Editor Report · Acceptance letter]

15 Aug 2024

PONE-D-24-10406R1 

PLOS ONE

Dear Dr. Groten, 

I'm pleased to inform you that your manuscript has been deemed suitable for publication in PLOS ONE. Congratulations! Your manuscript is now being handed over to our production team.

Kind regards, 

on behalf of

Dr. Edina Cenko 

Academic Editor

PLOS ONE